# Study Protocol for Two-Steps Parallel Randomized Controlled Trial: Pre-Clinical Usability Tests for a New Double-Chamber Syringe

**DOI:** 10.3390/ijerph17228376

**Published:** 2020-11-12

**Authors:** Pedro Parreira, Liliana B. Sousa, Inês A. Marques, Paulo Santos-Costa, Sara Cortez, Filipa Carneiro, Arménio Cruz, Anabela Salgueiro-Oliveira

**Affiliations:** 1The Health Sciences Research Unit, Nursing, Nursing School of Coimbra, 3004-011 Coimbra, Portugal; parreira@esenfc.pt (P.P.); ines.marques@student.uc.pt (I.A.M.); paulocosta@esenfc.pt (P.S.-C.); acruz@esenfc.pt (A.C.); anabela@esenfc.pt (A.S.-O.); 2Biophysics Institute, Coimbra Institute for Clinical and Biomedical Research (iCBR) Area of CIMAGO, Faculty of Medicine, CIBB, University of Coimbra, 3000-354 Coimbra, Portugal; 3Muroplás—Plastic Engineering Industry, 4745-334 Muro, Portugal; sara.cortez@muroplas.pt; 4PIEP—Innovation in Polymer Engineering, Guimarães, 4800-058 Braga, Portugal; f.carneiro@piep.pt

**Keywords:** usability tests, medical devices, nursing research

## Abstract

A new double-chamber syringe (DUO Syringe) was developed for intravenous drug administration and catheter flushing. This study presents a protocol for pre-clinical usability tests to validate the golden prototype of this new device, performed in a high-fidelity simulation lab by nurses. A two-steps parallel randomized controlled trial with two arms was designed (with standard syringes currently used in clinical practice and with the DUO Syringe). After randomization, eligible and consented participants will be requested to perform, individually, intravenous drug administration and flushing, following the arm that has been allocated. The procedure will be video-recorded for posterior analyses. After the completion of the tasks, nurses will be asked to answer a demographic survey, as well as an interview about their qualitative assessment of the device. A final focus group with all participants will also be conducted. Primary outcomes will concern the DUO Syringe’s effectiveness, efficiency, and safety, while secondary outcomes will focus on nurses’ satisfaction and intention of use. The pre-clinical protocol was defined according to the legal requirements and ISO norms and was reviewed and approved by the Ethics Committee of the Health Sciences Research Unit: Nursing of the Nursing School of Coimbra.

## 1. Introduction

Peripheral intravenous catheterization is the most common medical procedure performed by nurses, enabling the administration of intravenous solutions. This invasive practice is frequently associated with several complications (e.g., phlebitis, bloodstream infection, dislodgment, obstruction), leading to premature catheter replacement. Peripheral intravenous catheter (PIVC) related complications increase the length of hospital stay and, subsequently, healthcare costs [1]. Current complication rates can be decreased through preventive practices, especially PIVC flushing. This technique consists of rinsing the PIVC before and after its use [2]. Flushing not only maintains PIVC patency but also prevents the mixing of incompatible drugs/solutions [2]. Several studies showed that PIVC flushing removes blood and drug sediments from the catheter’s lumen, which potentiates catheter obstruction and bacterial colonization [3,4,5]. Considering international recommendations, to accurately administer intravenous drugs, nurses should flush the catheter pre, post, and between each drug administration, requiring at least two syringes. Despite the high level of policy awareness, nurses do not always follow international best practice recommendations regarding flushing practices [6,7]. To overcome this, prefilled flush syringes and multi-chamber syringes have been developed and distributed in international markets. Nevertheless, the use of prefilled syringes across international settings is minimal [8] and the need to use the same two syringes remains. Some double-chamber syringes were developed for drug reconstitution to promote faster drug administration by presenting a chamber with a prefilled drug in liquid or lyophilized form while maintaining sterility [9]. Others enable the delivery of two fluids in two moments, with a prefilled flush solution chamber and an empty chamber to be filled with the drug component [10]. However, current double-chamber syringes do not accomplish the international guidelines on infusion nursing, because they only enable the administration of the flush solution after drug delivery (the flush solution chamber can only be used after the drug chamber is empty) and do not accomplish the patency assessment of the PIVC [11].

Facing this, the goal of our project is to develop a new medical device that allows nurses to conduct intravenous drug administration and catheter flushing using only one double-chamber syringe. This new device is a double-chamber syringe that can be filled with a flush solution and intravenous drug, without mixing them, and allow for the delivery of each fluid separately (in equal or different volumes), enabling the assessment of the catheter patency through the administration of flush solution, drug delivery, and a final flushing of the catheter using the remaining flush solution.

The development of medical devices has vastly increased over the last decade, presenting an important role in clinical practice by improving patients’ well-being and quality of life [12]. Given the insurgence of new devices, the transition between prototype development and its implementation into clinical practice requires strict assessment and regulation. Contrasting pharmaceuticals, medical devices do not have straight or explicit regulations to follow [13], with regulatory directives diverging between countries. To bridge this gap, an international volunteer group of regulators was created (International Medical Device Regulators Forum; IMDRF), based on the work of the Global Harmonization Task Force on Medical Devices (GHTF) [14], specifically to uniform the regulatory directives around the world. In the European Union (EU), a new Regulation 2017/745 (5 April 2017) was recently approved by the European Parliament and the Council on medical devices [15]. Moreover, the development and implementation of new devices must be conducted under established standards, defined, and published by the International Organization for Standardization (ISO), striving for product quality, safety, and efficiency [16,17,18,19,20,21]. The development of international recommendations highlights the need to integrate the Health Technology Assessment (HTA) core assumptions in the development of new medical devices. In fact, 2.5% of all security events in NSW Health had a medical device as the main responsible cause. In 2014, 4000 incidents were reported (from near-miss until the death of the patient), involving four categories of high-risk devices: volume infusion pumps, infusion syringes, patient-controlled analgesia pumps, and defibrillators. More than 80% of the clinical incidents involving these devices can be attributed to errors in use requiring usability test [22,23,24,25].

Usability is defined in ANSI/AAMI/IEC 62366-1 as “the extent to which a user can use a product to achieve goals with effectiveness, efficiency, and satisfaction in a specific context” [19]. Notwithstanding, the technical report AAMI/IEC/TIR 62366-2 presents a broader focus of usability, highlighting the importance of assessing medical devices’ task accuracy, completeness, and efficiency, as well as user satisfaction [20]. According to these international directives, the medical devices’ development process comprises several strict and sequential stages, to predict potential errors carried out by end users and prevent the risk of use error due to the ergonomic features of the device. The pre-clinical validation of several parameters related to ergonomic, usability, and human factors is required to ensure that the prototype meets all the requirements to move forward to clinical studies [17,18]. Thus, the purpose of this study is to present the usability validation protocol for a double-chamber syringe supported in the user-centered design (UCD) method [26,27,28]. Specifically, we aim to describe the pre-clinical validation steps that will be used during the assessment of the functional prototype to accomplish the requirements established by the regulatory entities. This study protocol was developed under the SPIRIT 2013 Guidelines [29], with some adjustments according to pre-clinical trials in healthcare simulation research specificities [30,31].

## 2. Materials and Methods

### 2.1. Study Design

A two-steps parallel randomized controlled trial (RCT, Figure 1) with two arms was designed (involving standard syringes currently used in clinical practice and the new double-chamber syringe), following the Consolidated Standards of Reporting Trials (CONSORT) 2010 recommendations [28,29,30]. Considering that this project comprises the development of a new medical device, the EU directives [15], implemented in Portugal by the Portuguese National Authority of Medicines and Health Products (INFARMED), and ISO Standards [16,17,18,19,20] were considered. The pre-clinical study is planned to last 3 months from the start of the recruitment process (in July 2020).

### 2.2. Participants Recruitment and Sample Size

The research team will send an invitation to the intended users for this medical device, namely nurses from local tertiary hospitals to participate in the study. All interested nurses will be screened for eligibility criteria (Table 1). Regarding sample size, around 15–25 participants are usually considered to enroll in the usability tests, 15 being the acceptable minimum number according to the regulatory entities of the USA [26]. Although in the EU these parameters are not well-established, the U.S. Food and Drug Administration’s (FDA) orientation guidelines for medical devices highlight the need to balance the heterogeneity and homogeneity of the samples, reflecting as much as possible the target population [27]. For the usability tests with the double-chamber syringe, the AAMI/IEC/TIR 62366-2 was used to accomplish the sample size determination. According to that standard, there are diminishing returns on detecting new usability problems when sample size goes beyond 10 for each distinct user group. This is based on calculations determining cumulative probability of detecting a usability problem: R = 1 − (1 − P)^n^, where R equals the cumulative probability of detecting a usability problem, P equals the probability of a single test showing a usability problem, and n equals number of participants [20]. Therefore, 10 nurses will be enrolled in each arm (20 for each phase of this study), in a total of 40 participants in overall pre-clinical usability testing. The participants will not be compensated for participation.

### 2.3. Patient and Public Involvement

No patients or participants from the general public will be involved in this trial.

### 2.4. Randomization and Blinding

Eligible and consenting participants will be randomly allocated to one of two arms (Figure 1) for each phase of the study, using simple randomization (1:1 ratio). Random allocations through random number sequences will be computer-generated through an online registration system (Sealed Envelope^TM^, London, England). To ensure allocation concealment, the study staff will randomize each participant unknowingly and informed the principal investigator. Considering the specificities of this device, both of the two steps of the RCT will not be blinded, since it consists of the use of different devices to conduct the experiments of each allocated arm. Despite this, the participants will not be directly informed of their group allocation and will only be informed about the broad purposes of the research and the specific tasks they need to accomplish in the usability test.

### 2.5. Materials

The usability tests will be conducted in a laboratory environment that simulates two areas commonly found in hospital wards: (i) a fully equipped treatment room, where nurses commonly prepare intravenous drugs; (ii) a fully equipped single-bed patient room. Both areas contain audio/video recording equipment, allowing for a panel of evaluators to follow the tests in real time, without direct interaction with the participants, and for posterior analyses. The tests will be performed using an upper arm simulator that allows for PIVC insertion and drug administration (Multi-Venous IV Training Arm Kit; Nasco Healthcare, Fort Atkinson, WI, USA). This is a lifelike adult arm reproduction with a multi-vein system designed for peripheral intravenous therapy in the antecubital fossa or dorsum of the hand, with median, basilic, and cephalic accessible and palpable veins that allow intravenous peripheral therapy simulation (Figure 2). The study protocol ensures that all participants received standardized instructions and materials to perform the usability test.

The double-chamber syringe was developed with a 20 mL size: 10 mL for drug chamber (blue plunger) and 10 mL for flush solution chamber (white plunger). This innovative double-chamber syringe enables the filling (none of the chambers are pre-filled) and administration of both solutions (verification of the catheter patency through the administration of the flush solution, the subsequent drugs delivery, and the final flushing of the catheter using the remaining flush solution) [32]. With the double-chamber syringe, the instructions for use will be provided to the participants, which will be also assessed by the participants in the interviews/focus groups.

Appendix A will be used, such as (i) the informed consent document, with a brief description of the study, the main purpose of the groups, and the voluntary nature of participation; (ii) non-disclosure agreement (NDA); (iii) anonymous demographic questionnaire to sample characterization; (iv) interviews and focus groups guidelines (Table 2); and (v) usability questionnaire (42 items in a 7-point Likert scale focused on significant usability dimensions: usefulness, ease of use, ease of learning, satisfaction, and intention to use).

### 2.6. Interventions

This pre-clinical validation will be assessed through a two-arm parallel RCT in two phases (Figure 1), in which the nurses will be requested to perform intravenous drug administration per the allocated arm in the simulated setting. In order to ensure the standardized conditions for the procedure, the PIVC is already in place (being placed always by the same nurse of the research team). In the first phase (intra-subject analyses), the 20 nurses will perform the protocol using the two types of syringes, starting with the double-chamber syringe and, then, a standard syringe (arm A), or in reversed order (arm B). In the second phase (inter-subject analysis), another 20 nurses (without any previous information or contact with this double-chamber syringe) will be allocated to perform the protocol with a double-chamber syringe (arm A) or with standard syringes (arm B). The usability test session will be run as a single participant test, with each participant repeating the procedure three times, under external supervision. The procedure is divided in two main phases (drug preparation and drug administration), and tasks were defined in accordance to both study groups (Table 3 and Table 4). The procedure will be recorded by audio/video for posterior analyses. At the end of the usability tests, interviews/focus group with all participants will be conducted to collect detailed qualitative information about this new medical device.

### 2.7. Outcomes

These usability tests are intended to measure primary (effectiveness, efficiency, and safety) and secondary outcomes (satisfaction and intention of use), ensuring that the medical device accomplishes the needed legal requirements before testing in real clinical settings. Specifically, in this pre-clinical study, the primary outcomes that will be measured are: (i) number of tasks and goals achieved; (ii) procedure execution time; and (iii) number, type, and intensity of errors (Table 5). These parameters will be extracted from the video recordings (usability tests) and compared to a detailed procedure checklist. The secondary outcomes will be assessed through the usability questionnaire and audio recordings from the individual interviews and focus group carried. As the participants perform each procedure three times (with the double-chamber syringe and standard syringes), the research team can also analyze their learning curve and training needs required to use this new medical device, along with the qualitative (interviews and focus group) and quantitative (usability questionnaire).

### 2.8. Data Collection, Management, and Analysis

Study participants identification numbers (ID) will be used, and all data will be anonymized for subsequent analysis and reports/publications. Individual information to be collected includes demographic (gender, age), academic qualifications (degree), and professional data (clinical experience, work setting) of the nurses eligible to perform the usability tests. The names of the participants on the consent forms will be stored separately in locked cabinets accessible only by named personnel.

Statistical analysis of the collected data will be performed using the Statistical Package for the Social Sciences, version 24 (IBM SPSS Statistics 24; SPSS Inc., Chicago, IL, USA). Means, standard deviations, frequencies, and percentages will be used as descriptive statistics (or median values and interquartile ranges for skewed data). The outcomes in the two groups in each study phase will be examined to detect the effect of group allocation through inferential statistics (Student’s t-test for independent and paired samples, or non-parametric equivalents, Mann-Whitney U and Wilcoxon tests; X^2^ test or Fisher’s exact test), considering a statistical significance level of 0.05 (two-sided significance level of 5%). Qualitative content analysis will also be conducted after the transcription of the individual interviews and focus group.

### 2.9. Ethical Considerations

This double-chamber syringe is a new medical device that is currently under development. Due to this, this pre-clinical protocol was defined according to the legal requirements of the European Union [15], and the ISO norms related to ergonomics and usability assessment of MDs [16,17,18,19,20,21]. The study protocol and all the templates that will be used in usability tests were reviewed and approved by the Ethics Committee of the Health Sciences Research Unit: Nursing of the Nursing School of Coimbra (Number P608-8/2019).

The eligible nurses will receive written and oral information about the study and written informed consent and a non-disclosure agreement (NDA) will be requested to the nurses that are willing to participate. As previously stated, participants will be assigned to an ID number, which will be used in all data collection instruments. Personal information will be separated from the main data collected and will not be shared. All the documentation related to the study will be saved in locked cabinets only accessible by the study staff. All collected data will be used exclusively for this study, and the confidentiality of participants will always be maintained. Participants have the right to withdraw from the study at any time, as established by national legislation, without any consequences for them. In this case, information related to the reason of withdrawing will be collected.

### 2.10. Dissemination

Due to the absence of specific guidelines for pre-clinical studies with medical devices, upon completion of the usability tests in both phases, the Consolidated Standards of Reporting Trials guidelines (CONSORT 2010) [28,29] will be used to report the data obtained, with adjustments for health care simulation research specificities [30,31]. All the data will not be publicly available but will be accessible from the principal investigator on reasonable request. The pre-clinical research results will be disseminated open-access, peer-reviewed journals and national and international scientific meetings. Authorship will be considered according to the recommendations of the International Committee of Medical Journal Editors [33], regarding the contributions to the design, conduct, interpretation, and reporting the pre-clinical data [34].

## 3. Results

Pre-clinical studies on this new double-chamber syringe will be initiated soon, with the recruitment of eligible nurses to perform the usability tests in a simulation lab. The clinical trial is already registered at ClinicalTrials.gov (NCT04046770). After the pre-clinical studies, all legal documentation will be submitted to Portuguese legal entities to implement the clinical trial in real hospital setting.

## 4. Discussion

Peripheral intravenous catheterization is the most frequent invasive procedure performed in nursing clinical practice and enables the intravenous administration of fluids directly on the bloodstream [35,36]. In this clinical practice, there is a wide range of complications that can impact patient´s safety [37,38], and the flushing procedure is recommended to prevent major mechanical, vascular, or infectious complications [39,40]. In several international guidelines on infusion nursing flushing is also mentioned to assess the catheter patency previously to drug administration and to clean the catheter after the drug delivery [2,3,41]. The main purpose of flushing is to maintain PIVC patency by preventing internal luminal occlusion, reducing the build-up of blood or other products on the catheter internal surface, also preventing interaction between incompatible fluids or drugs [5,42,43].

Currently, this process requires at least the use of two different syringes (one for drug and one for flush solution). The double-chamber syringe will enable nurses to perform the recommended pre- and post-PIVC flush during intravenous therapy administration, reducing the catheter manipulations and subsequent possible complications. Moreover, the authors expect that this new medical device can assist healthcare organizations in providing sustainable care, by reducing the number of syringes needed for such a recurrent procedure.

The development of this new device has followed the UCD method, widely recognized as the gold standard to address the real needs of the targeted end-users. Given its potential impact on patients’ safety and well-being, medical devices require safety and usability testing before market distribution [25]. Usability tests are universally accepted as essential requirements for the validation of devices’ effectiveness, efficiency, safety, and satisfaction parameters [24]. Although medical devices’ regulation is not yet unvarying, the IMDRF directives, along with the EUnetHTA JA2 2015, transposed some methods from pharmaceuticals regulation (e.g., effectiveness of the new medical device compared with the standard method/device) and provided international guidance [13,44,45] that were used to define this pre-clinical study protocol.

## 5. Conclusions

Following international legal directives, a protocol for pre-clinical usability tests was delineated, comparing the standard syringes currently used in clinical practice and the new double-chamber syringe. The authors expect that the development of this new medical device (double-chamber syringe) will improve nursing practice by enhancing adherence to quality and safety recommendations in the administration of intravenous therapy.

## Figures and Tables

**Figure 1 ijerph-17-08376-f001:**
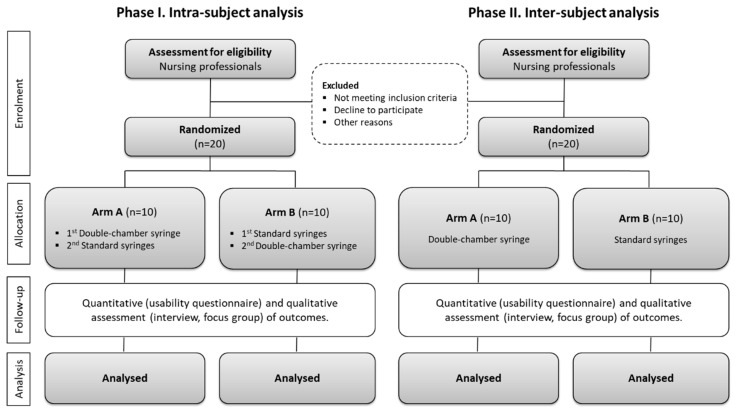
Flowchart of the two-arm parallel randomized controlled trial (RCT).

**Figure 2 ijerph-17-08376-f002:**
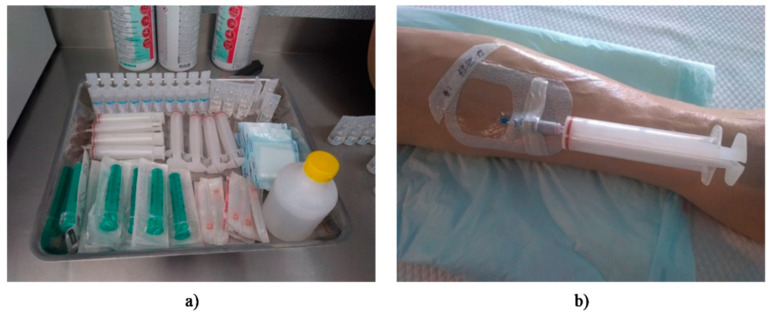
Simulation environment in nursing laboratory for usability tests, with (**a**) the material available for the participants and (**b**) the training arm kit with an inserted peripheral intravenous catheter (PIVC) to simulate intravenous drug administration and flushing.

**Table 1 ijerph-17-08376-t001:** Eligibility criteria for participants.

Inclusion Criteria	Exclusion Criteria
✓ Nurses	✓ Other health professionals
✓ Minimum academic qualifications of a bachelor degree	✓ Any previous contact with the device (knowledge and/or manipulation)
✓ Experience on intravenous drug administration	✓ Individuals who have a financial relationship with the device manufacturer and/or distributor

**Table 2 ijerph-17-08376-t002:** Interviews and focus groups guidelines.

Steps	Duration (Min)	Purposes
Introduction	5 min	✓ Presentation of the main objective for the interview/focus group.
Discussion	30 min	✓ Discussion on the main advantages and limitations of the new double-chamber syringe, comparing them with the standard syringes (“We would like you to summarize your experience with this new medical device, the Duo Syringe. We want you to help us to identify its advantages and potential, as well as any limitations or difficulties that you have identified when using it in these usability tests”). ✓ The participants are also asked to answer the usability questionnaire.
Conclusion	5 min	✓ Show appreciation for participation and obtain main (anonymous) information to characterize the group.

**Table 3 ijerph-17-08376-t003:** Procedures to be accomplished by the study participants: drug preparation.

	Standard Syringe	Double-Chamber Syringe
Phase 1 (Drug Preparation)	Determine the total drug dose to be administered according to chart prescription. Check the drug ampule for intactness, cloudiness, particles, and color. Gently flick or tap the top of the ampule to remove medication trapped in the top of the ampule. Wrap a 2 in. × 2 in. gauze pad around the neck of the ampule, snap the top off, breaking it. Discard the top in a sharps container. Attach a 18G filter needle to a 10 mL syringe. Withdraw the medication from the ampule. Hold the syringe vertically and draw 0.2 mL of air into the syringe. Remove the filter needle and attach a blunt 18G needle. Eject the 0.2 mL of air and read the dose. Attach a blunt 18G needle to a 10 mL syringe. Remove the neck of a plastic ampule containing 0.9% normal saline for catheter flush. Discard the neck. Withdraw 10 mL of normal saline from the ampule. Hold the syringe vertically and draw 0.2 mL of air into the syringe. Eject the 0.2 mL of air and read the dose. Label both syringes. Prepare a tray with the syringes, pair of clean gloves, sterile gauze, and a 2% chlorhexidine gluconate in 70% isopropyl alcohol spray.	Determine the total drug dose to be administered according to chart prescription. Attach a 18G filter needle to the syringe. Remove the neck of a plastic ampule containing 0.9% normal saline for catheter flush. Discard the neck. Withdraw 10 mL of normal saline from the ampule to the white plunger chamber. Hold the syringe vertically and draw 0.2 mL of air into the syringe. Eject the 0.2 mL of air and read the dose. Check the drug ampule for intactness, cloudiness, particles, and color. Gently flick or tap the top of the ampule to remove medication trapped in the top of the ampule. Wrap a 2 in. × 2 in. gauze pad around the neck of the ampule, snap the top off, breaking it. Discard the top in a sharps container. Withdraw the medication from the ampule to the blue plunger chamber. Hold the syringe vertically and draw 0.2 mL of air into the syringe. Remove the filter needle and attach a blunt 18G needle. Eject the 0.2 mL of air and read the dose. Label the syringe. Prepare a tray with the double-chamber syringe, pair of clean gloves, sterile gauze, and a 2% chlorhexidine gluconate in 70% isopropyl alcohol spray.

**Table 4 ijerph-17-08376-t004:** Procedures to be accomplished by the study participants: drug administration.

	Standard Syringe	Double-Chamber Syringe
Phase 2 (Drug administration)	Don clean gloves. Disinfect the catheter hub with the 2% chlorhexidine gluconate in 70% isopropyl alcohol spray. Place a sterile gauze below the catheter hub. Holding the flushing syringe, remove both the needle and the cap, and place them on a sterile surface. Insert the flush syringe into the catheter hub. Administer a 5 mL flush to clear the line with a push–pause technique. Continue to hold the catheter hub and apply positive pressure on the plunger. Remove the flush syringe, attach the needle, and place it on a sterile surface. Insert the drug syringe into the catheter hub. Administer the prescribed drug. Continue to hold the catheter hub and apply positive pressure on the plunger. Remove the drug syringe. Holding the flushing syringe, remove both the needle and the cap. Insert the flush syringe into the catheter hub. Administer a 5 mL flush to clear the line with a push–pause technique. Remove the flush syringe. Remove the sterile gauze. Leave the patient room.	Don clean gloves. Disinfect the catheter hub with the 2% chlorhexidine gluconate in 70% isopropyl alcohol spray. Place a sterile gauze below the catheter hub. Holding the flushing syringe, remove both the needle and the cap, and place them on a sterile surface. Insert the double-change syringe into the catheter hub. Administer a 5 mL flush to clear it with a push–pause technique. Continue to hold the catheter hub. Apply positive pressure on the drug chamber’s plunger and administer the prescribed dose. Continue to hold the catheter hub. Apply positive pressure on the flushing chamber’s plunger. Administer a 5 mL flush to clear the line with a push–pause technique. Remove the double-chamber syringe. Remove the sterile gauze. Leave the patient room.

**Table 5 ijerph-17-08376-t005:** Types of errors to be assessed during the study.

Phase 1 (Drug Preparation)	Phase 2 (Drug Administration)
✓ Defective connection of the needle to the syringe (both groups); ✓ Non-compliance with aseptic technique during needle connection (both groups); ✓ Incorrect drug dose or flushing volume after aspiration (in mL) (both groups); ✓ Non-compliance with aseptic technique during drug/flushing aspiration (both groups); ✓ Incorrect syringe labelling or label position (both groups); ✓ Wrong chamber aspiration sequence (double-chamber group); ✓ Aspiration of drug/flushing solution to the wrong chamber (double-chamber group); ✓ Chamber contamination (double-chamber group).	✓ Defective connection of the syringe to the catheter hub (both groups); ✓ Non-compliance with aseptic technique during syringe-to-hub connection (both groups); ✓ Incorrect administration of drug dose or flushing volume (in mL) (both groups); ✓ Non-compliance with aseptic technique during drug/flushing administration (both groups); ✓ Wrong administration sequence (both groups); ✓ Non-compliance with the push–pause technique during catheter flushing (both groups); ✓ Chamber contamination (double-chamber group); ✓ Catheter accidental removal (both groups).

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
