# Peer review of "Study Protocol for Two-Steps Parallel Randomized Controlled Trial: Pre-Clinical Usability Tests for a New Double-Chamber Syringe"

_ijerph, 2020, doi:10.3390/ijerph17228376_

Round 1

Reviewer 1 Report

This paper described a usability validation study protocol for a novel double-chamber syringe.

In general, this paper is acceptable. However, the author needs to address below concerns from the reviewer:

  1. The reference to ISO 62366 is incorrect. Please reference the ANSI/AAMI/IEC 62366 instead. 

  2. The background of this protocol is almost the same narrative with the clinical study protocol that has already been published (https://www.ncbi.nlm.nih.gov/pmc/articles/PMC6961373/). Although issues may overlapping, the usability testing should have different background than the clinical study. Please refer to the IEC 62366 for prerequisites of conducting a usability validation plan, such as defining intended user, intended use environment, user interface characteristics relating to safety and potential user errors, summary of known user problems, and etc.

  3. Table 1: The inclusion criteria includes "experience on intravenous drug administration". As the experience level of user may significantly affect the effectiveness and safety of the device, this inclusion criteria dose not address this confounder for your study design. Please specify why the level of experience of the user is not considered in your protocol.
  4. The intended user for this device was not clearly stated in the protocol. Please specify.
  5. It is unclear if the intended user will need training for using the device. Please clarify.

  6. Line 117-124: you presented a formula to determine the sample size. However, it is unclear how the formula supported your sample size of 10 for each arm. Please specify.

  7. For the secondary outcome, you will use survey or focused group to obtain qualitative data. Please provide the survey and focused group design and questions for the completeness of the protocol.

Author Response

The authors would like to thank Reviewer 1 for the time spent in the thorough analysis of the manuscript and recommendations made to improve it. Below are our answers:

Reviewer 1: The reference to ISO 62366 is incorrect. Please reference the ANSI/AAMI/IEC 62366 instead. 

Authors: Thank you, this was a clear mistake on our end. The reference was corrected.

Reviewer 1: The background of this protocol is almost the same narrative with the clinical study protocol that has already been published (https://www.ncbi.nlm.nih.gov/pmc/articles/PMC6961373/). Although issues may overlapping, the usability testing should have different background than the clinical study. Please refer to the IEC 62366 for prerequisites of conducting a usability validation plan, such as defining intended user, intended use environment, user interface characteristics relating to safety and potential user errors, summary of known user problems, and etc.

Authors: We appreciate Reviewer 1 feedback concerning the background section. While we understand that the initial section of the background is similar to the clinical trial protocol, this is due to the problem being the same (lack of standardized flushing practices and negative outcomes that can occur). However, the remaining section (lines 57-97) are focused on usability testing. We appreciate Reviewer 1 suggestion and have included a mention to IEC 62366.

Reviewer 1: The intended user for this device was not clearly stated in the protocol. Please specify.

Authors: We appreciate this comment and have now clarified the intended user in the Participants recruitment and sample size section.

Reviewer 1: It is unclear if the intended user will need training for using the device. Please clarify. 

Author: Thank you for this suggestion! We have clarified this in the outcomes section. 

Reviewer 1: Line 117-124: you presented a formula to determine the sample size. However, it is unclear how the formula supported your sample size of 10 for each arm. Please specify.

Authors: Thank you! We have now clarified sample size determination. According to AAMI/IEC/TIR 62366-2, 10 participants in each group are enough to detect new usability problems. Therefore, 10 nurses will be enrolled in each arm (20 for each phase of this study), in a total of 40 participants in overall pre-clinical usability testing.

Reviewer 1: For the secondary outcome, you will use survey or focused group to obtain qualitative data. Please provide the survey and focused group design and questions for the completeness of the protocol.

Authors: The main guidelines for the interviews and focus groups can now be found in Table 2, including the questions posed to the participants. As supplementary material, authors provide the usability questionnaire. Thank you for this suggestion!

Reviewer 2 Report

It's a very nice Study Protocol.

Normally for the minimized statistical analysis sample size is at least 30. Wonder why this protocol only choose 10 for participants per group only?

Some languages of structure can be better. For example, on page 1 line 40 "Flushing maintains PIVC patency but also prevents the mixing of incompatible drugs/solutions." is not clearly presenting to the reader. Maybe can change to "Flushing not only maintains PIVC patency but also prevents the mixing of incompatible drugs/solutions." 

Author Response

The authors would like to thank Reviewer 2 for the time spent on the revision of the manuscript and constructive feedback provided. Below are our answers per comment made:

Reviewer 2: Normally for the minimized statistical analysis sample size is at least 30. Wonder why this protocol only choose 10 for participants per group only?

Authors: According to AAMI/IEC/TIR 62366-2, which is an important standard on usability engineering application to medical devices, 10 participants in each group are enough to detect new usability problems. Therefore, 10 nurses will be enrolled in each arm (20 for each phase of this study), in a total of 40 participants in overall pre-clinical usability testing.

Reviewer 2: Some languages of structure can be better. For example, on page 1 line 40 "Flushing maintains PIVC patency but also prevents the mixing of incompatible drugs/solutions." is not clearly presenting to the reader. Maybe can change to "Flushing not only maintains PIVC patency but also prevents the mixing of incompatible drugs/solutions." 

Author: The authors would like to thank reviewer 2 for the thorough analysis of the manuscript and semantic/structural suggestions! All suggestions were integrated into the manuscript. Other minor revisions were made after a second analysis of the manuscript (highlighted in track changes). 

Reviewer 3 Report

The authors attempt to explain the sample size calculation, but it is difficult to understand.
They should explain in detail how they arrived at the conclusion that n=20 is statistically sufficient, and they should also provide specific values for the parameters used in their determination.
It also appears that n=20 is too small for a randomized controlled trial.

Author Response

The authors would like to thank Reviewer 3 for the time spent in the assessment of this manuscript and constructive feedback provided during its revision. Below are our answers to the comments made.

Reviewer 1: The authors attempt to explain the sample size calculation, but it is difficult to understand. They should explain in detail how they arrived at the conclusion that n=20 is statistically sufficient, and they should also provide specific values for the parameters used in their determination. It also appears that n=20 is too small for a randomized controlled trial.

Authors: According to AAMI/IEC/TIR 62366-2, which is an important standard on usability engineering application to medical devices, 10 participants in each group are enough to detect new usability problems. Therefore, 10 nurses will be enrolled in each arm (20 for each phase of this study), in a total of 40 participants in overall pre-clinical usability testing.

Round 2

Reviewer 1 Report

My questions have all been addressed adequately. This manuscript is acceptable for publish.